# Hazardous Child Labour, Psychosocial Functioning, and School Dropouts among Children in Bangladesh: A Cross-Sectional Analysis of UNICEF’s Multiple Indicator Cluster Surveys (MICS)

**DOI:** 10.3390/children10061021

**Published:** 2023-06-07

**Authors:** Aye Myat Thi, Cathy Zimmerman, Meghna Ranganathan

**Affiliations:** 1Innovations for Poverty Action, Yangon 11111, Myanmar; 2Department of Global Health and Development, London School of Hygiene and Tropical Medicine, London WC1H 9SH, UK

**Keywords:** hazardous child work, psychosocial functioning, school dropout, Bangladesh, low and middle-income countries

## Abstract

Child labour is a common financial coping strategy in poor households, especially in low-and middle-income countries with many children working under hazardous conditions. Little is known about the linkages between hazardous work conditions and psycho-social and educational outcomes. We analysed the Bangladesh Multiple Indicator Cluster Survey (BMICS) round 6 to assess the association between the exposure variables, including child labour, hazardous child labour (HZCL) and hazardous work, and outcome variables, including psychosocial functioning difficulty and school dropout, in children aged 5 to 17 years. We conducted bivariable and multivariable analyses to examine the association. In the adjusted analyses, children engaged in HZCL had increased odds of psychosocial functioning difficulty (aOR: 1.41; 95% CI: 1.16–1.72) and school dropout (aOR: 5.65; 95% CI: 4.83–6.61) among 5–14-year-olds compared to children who did not engage in child labour and hazardous work. Other independent factors associated with psychosocial functioning difficulty and school dropout included being male, living in a deprived neighbourhood, being exposed to violent punishment, the caregiver’s attitude towards physical punishment, the mother’s functional difficulty and lower maternal education. The linkages between hazardous work and psychosocial functioning difficulty appear more prominent among children not in school. Further, the evidence on the relationship between hazardous work and school dropout is stronger among children with psychosocial functioning difficulty. Policies and programmes that target the most hazardous forms of work are likely to have the greatest benefits for children’s mental health, social well-being and educational attainment.

## 1. Introduction

In 2020, approximately ten percent of children globally (160 million) were engaged in child labour [1], with almost half (79 million) being engaged in hazardous work [1]. Child labour threatens the health, development and education of children [1], although not all forms of work are harmful for children, and in fact some participation in work can be positive for young people [2]. However, hazardous work is associated with health risks, and is defined as “work which, by its nature or the circumstances in which it is carried out, is likely to harm the health, safety or morale of children” [3]. The largest number of children in hazardous work is found in the Asia-Pacific region [3]. Among children in hazardous work, 60% are boys and they commonly work in agriculture and manufacturing, while girls are frequently engaged in in service-oriented work [3].

Studies indicate that child labour poses numerous physical and psychological health risks and is associated with the negative effects of depriving children of an ordinary childhood [4]. Long work hours that are common in child labour leave little time for young people to socialise with peers and family members, which is fundamental to psychosocial development [5]. Failure to develop positive personal relationships may cause children to be less confident, less communicative, and more isolated, leaving them at increased risk of depression, emotional and behavioural problems [5]. A recent systematic review conducted in low- and middle-income countries (LMICs) also indicated that child labour is strongly associated with poor mental health outcomes [6]. Subsequently, emotional and behavioural problems associated with child labour can impair children’s capacity to learn and impose time constraints that hinder their participation in school activities [7,8], affecting both school performance and attendance. Population studies suggest that a higher prevalence of child labour is associated with low literacy rates and high rates of repeating grades and school dropout [9].

Children’s engagement in ‘hazardous work’ exacerbates the negative health and education impacts of child labour [10]. The term ‘hazardous work’, or harmful forms of work, includes working in dangerous situations, with dangerous machinery/equipment and being exposed to harmful substances [3]. While the most visible harm from child labour is physical, hazardous child labour results in numerous psychosocial consequences that can have long-term effects on children’s social engagement and education. However, compared to the physical effects of child labour, to date, there has been relatively little attention paid to the psychosocial implications [3]. Because emotional and social functioning often manifest independently of physical health, exposures, consequences, and the identification of interventions require a focused investigation [3,11].

A recent evidence review by Boutin et al. (2022) suggested that certain methodological approaches that focused narrowly on physical health or educational outcomes risked missing the interdependent nature of the effects, resulting in either overestimations or underestimations of impact [12]. Despite the evidence about the relationship between hazardous work and developmental and educational consequences [12], there remain relatively limited analyses on the multidimensional effects of hazardous work and how adverse consequences interact with psychosocial functioning and educational outcomes.

South Asia has an especially high number of working children, estimated at approximately 30 million, with a large proportion engaged in hazardous work and the worst forms of child labour [13,14]. Following India, Bangladesh has the second highest burden of child labour in South Asia [13,14]. Moreover, Bangladesh has the largest share (75%) of children in harmful work and the highest out of school children among 15–17-year-olds [13,15,16]. This paper aims to (1) examine hazardous work conditions and the association with psychosocial (cognitive, emotional and behavioural) functioning difficulty and school dropout rates in a high-burden context for hazardous work; and (2) investigate the differential effects of hazardous work conditions by psychosocial functioning difficulty or school dropouts in the subgroup analyses. Developing an understanding of the different effects of hazardous child labour can inform tailored interventions for vulnerable children to ensure equitable life opportunities. This analysis of the 2019 Bangladesh Multiple Indicator Cluster Survey (BMICS) round 6 survey data on children aged 5–17 in Bangladesh, adds to the evidence base by examining the complex interaction between hazardous child labour, school dropout, the psychological adjustment of children and parental behaviours.

## 2. Materials and Methods

### 2.1. Study Design and Setting

The present study is a secondary data analysis of the BMICS round 6 survey. The BMICS was carried out in 2019 by the Bangladesh Bureau of Statistics (BBS) in collaboration with UNICEF Bangladesh [17]. The survey is primarily designed to provide internationally comparable national-level data with information on socio-demographic characteristics and health of children, adolescents, women and households in Bangladesh [17]. The 2019 BMICS employed a two-stage stratified cluster sampling approach that was based on the 2011 Bangladesh Census of Population and Housing (2011). The sampling strata extended to urban and rural areas within 64 districts in 8 divisions of the country (Barishal, Chattogram, Dhaka, Khulna, Mymensingh, Rajshahi, Rangpur and Sylhet) [17]. Within each stratum, specified census enumeration areas (EAs), known as the primary sampling units (PSUs), were selected systematically with probability proportional to size at the first stage. Within each EA, a listing of households was carried out and a sample of households was selected at the second stage. Data were collected from 64,400 eligible households between January and June 2019 [17].

### 2.2. Measures, Variables and Definitions

We used data from the ‘mother/primary caretakers of children 5–17 years’ and used this dataset to create the exposure, outcome and confounding variables. Variables such as age, sex, highest education level of the child, education level of the mother/caretaker, wealth index quintile, area of the household, mother’s/caregiver’s attitudes towards physical punishment, and functional difficulties of primary caretaker were also included in the analysis. We created new variables for the exposure variable child labour, and outcome variables psychosocial functioning difficulty, and school dropout. Please see Table 1 for the operational definitions of variables. Child labour was measured by asking mothers or primary caretakers of children aged 5–17 years about one randomly selected child in each household to see whether the child performed economic activities. Children who were not attending school at the time of the survey such as those in pre-school or in early childhood education, such as nursery (*n* = 253), were excluded from this classification.

#### Confounding Variables

We identified confounding variables based on the literature and conceptual framework (see below). These variables were categorised based on the 2019 BMICS [17]: age (5–11 years, 12–14 years, 15–17 years), sex (female, male), education level of the child (ECE, primary, secondary, secondary/higher secondary and higher education), wealth index quintile (poorest, second, middle, fourth, richest), area of the household (urban, rural), mother’s/caretaker’s educational level (pre-primary/none, primary, secondary and higher secondary and above), mother/caretaker’s attitudes towards physical punishment (yes/no), functional difficulties of primary caretaker (yes/no) and child discipline (violent discipline, non-violent discipline and no discipline) [17].

### 2.3. Conceptual Framework

Informed by the literature and the variables classified above, we developed a conceptual framework to guide the analysis (please see Figure 1). This framework aims to depict the links between hazardous child labour and psychosocial functioning and school dropout. The framework also accounts for potential confounders, including age, sex of the child, socio-economic status of the family, child’s and mother’s/guardian’s education, mother’s/guardian’s functional difficulties, exposure to harsh punishment and the guardian’s attitudes towards physical punishment.

### 2.4. Statistical Analysis

From the 2019 BMICS survey data, the SPSS data file of ‘mother/primary caretakers of children 5–17—fs.sav’ data were imported into Stata/SE 17 for Windows. First, we undertook descriptive statistical analyses to characterise the study population. As data were collected for a randomly selected child in the households with at least one child aged 5–17, the respective sample weight (fsweight) is used to generate estimates of exposures and outcome variables accounting for random selection.

First, we used logistic regression, as the outcome variables (psychosocial functioning difficulty and school dropouts) are categorical, and the exposure variables are a mixture of both continuous and categorical [18]. Second, to assess the relationship between independent and dependent variables, bivariable analyses were performed using χ^2^ tests. Then, multivariable logistic regression analyses were adjusted for potential confounding variables to determine the relationship between hazardous child labour and psychosocial functioning difficulty, and then the relationship between hazardous child labour and school dropout. These multivariable analyses were run for children aged 5–14 years, given that child discipline variables were available for this age range. Variables that showed collinearity were dropped from multivariable regressions.

Finally, subgroup analyses investigated associations between hazardous work and psychosocial functioning difficulty by status of school dropout, and associations between hazardous work and school dropout by psychosocial functioning difficulty. The level of statistical significance was set as a two-sided *p* value ≤ 0.05 and analyses were calculated for both weighted percentages. Records with missing functioning variables (*n* = 1110) were removed from the dataset. Then, children who were in pre-school or non-standard school or early childhood education (*n* = 253) were excluded from the school dropout variable.

## 3. Results

### 3.1. Characteristics of the Study Population

Table 2 shows the socio-demographic characteristics of children in Bangladesh. Of 66,705 children aged 5–17 years, most were from rural areas (79.5) and approximately half were between the ages of 5 and 11 and had attained primary education. The findings indicate that 6.7% of children were in child labour, 7.9% worked in hazardous conditions and 3.4% were in hazardous child labour. Nearly all children (92.7%) were subjected to violent discipline (physical punishment or psychological aggression) based on the study definition by MICS, among whom about one-third (35.1%) of their mother’s/guardian’s agreed that physical punishment was the proper way of raising a child. Just over ten percent (11.8%) of all children had dropped out of school.

### 3.2. Hazardous Child Labour and Psychosocial Functioning Difficulty in Children

The proportion of children with psychosocial functioning difficulty was highest among children in hazardous child labour (11%), followed by 9.1% among those in child labour, 8.5% among those working under hazardous conditions and 7.2% among those not in child labour.

Hazardous child labour (aOR: 1.41; 95% CI: 1.16–1.72; P: 0.001) and hazardous work (aOR: 1.39; 95% CI: 1.1–1.76; P: 0.006) were associated with psychosocial functioning difficulty after adjusting for confounders in 5–14-year-olds. Poor psychosocial functioning was associated with violent discipline (aOR: 1.73; 95% CI: 1.43–2.10), school dropout (aOR: 1.63; 95% CI: 1.39–1.92) and being from rural areas (aOR: 1.25; 95% CI: 1.11–1.42). Children aged 5–14 years old were less likely to have psychosocial difficulties if they were female, from a more affluent neighbourhood and if their parent/guardian did not endorse physical punishment and their mothers had no functional difficulty (aOR: 0.26; 95% CI: 0.22–0.30) (please see Table 3).

Associations between hazardous child labour and psychosocial functioning difficulty were assessed in the school dropouts and school-attending groups and accounted for similar confounding factors to those in Table 3. Table 4 shows that hazardous child labour was significantly associated with psychosocial functioning difficulty in both groups. However, the effects of hazardous work appeared to differ by schooling status. Among the children who dropped out of school, children in hazardous work (without child labour) were more likely to have psychosocial functioning difficulty (aOR: 2.55; 95% CI: 1.61–4.06; P: 0.000), while this association was not striking among respective counterparts among children who attended school.

### 3.3. Hazardous Child Labour and School Dropout in Children

The proportion of those that dropped out from school was 54.3% among those exposed to hazardous child labour and 34.3% among those working under hazardous conditions, 30% among those in child labour and 8.4% among those not in child labour.

Multivariable analysis showed that compared to children not in child labour and hazardous work, children engaged in child labour (aOR: 3.97; 95% CI: 3.38–4.67), hazardous child labour (aOR: 5.65; 95% CI: 4.83–6.61) and hazardous work (aOR: 1.69; 95% CI: 1.38–2.06) were associated with dropping out of school. Furthermore, higher dropout rates were associated with psychosocial difficulties (aOR: 1.71; 95% CI: 1.46–1.99) and older age (aOR: 1.51; 95% CI: 1.47–1.54). Protective factors for school retention included being female, living in an affluent neighbourhood and being from a rural area. Children whose mothers had a higher secondary level or above were less likely to drop out (aOR: 0.21; 95% CI: 0.15–0.29). Interestingly, being exposed to harsh discipline was also linked to lower odds of school dropout (please see Table 5).

Subgroup analyses reported that different types of child labour were significantly associated with school dropout, irrespective of psychosocial functioning difficulty (Table 6), although the strengths of association differed in the two groups. Among children without psychosocial functioning difficulty, those engaged in child labour and hazardous child labour were four to six times as likely to drop out of school compared to those not in child labour or hazardous work. We found weaker associations among children with psychosocial functioning difficulty (aOR: 3.33 child labour; aOR: 4.5 hazardous child labour). In contrast, children in hazardous work (without child labour) were 2.5 times more likely to drop out of school in the group with psychosocial functioning difficulty. Meanwhile, the association between hazardous work and school dropout was less pronounced in the group without psychosocial functioning difficulty (aOR: 1.5).

## 4. Discussion

This secondary analysis of the MICS data, a nationally representative sample of children in Bangladesh, confirms the vital relationship between hazardous forms of child labour, psychosocial functioning and school dropout. Our results indicated that about 8% of children were working in hazardous conditions including 3.4% who were in conditions considered as hazardous child labour and 4.6% were in hazardous work (without child labour). Children in hazardous child labour were 1.4 times more likely to have psychosocial functioning difficulty and almost six times likely to drop out from school than children who were not in child labour or hazardous work. In school dropouts, there was evidence that hazardous child labour and hazardous work were associated with psychosocial functioning difficulty. Among the school-attending group, only children in hazardous child labour were more likely to have psychosocial functioning difficulty. Irrespective of psychosocial functioning difficulty, different types of child labour were associated with increased likelihoods of school dropout. Notably, children involved in hazardous work were 2.5 times more likely to drop out from school, in the group with psychosocial functioning difficulty, while their respective counterparts were 1.5 times more likely to drop out, in the group without psychosocial functioning difficulty.

These findings on the effects of child labour are not surprising and reflect the large body of literature on the harmful effects of child labour, including studies in Brazil [19], Ethiopia [20] and India [21], where there are reported associations between hazardous/child labour and poor psychosocial well-being. The current study adds to the evidence base by demonstrating that children working under hazardous work conditions with/without child labour have greater psychosocial functioning difficulty. This is consistent with a secondary analysis of the UNICEF MICS data from 15 countries which showed that children with disability, particularly those who had poor mental health or cognitive functioning, were more likely to be exposed to hazardous child labour [22]. Furthermore, a multi-national study from Afghanistan, Bangladesh, Nepal and Pakistan helps to decipher some of the psychosocial features affected by work among youth in the brick industry. Working children had lower cognitive abilities, a lower sense of control over life, more stress-related emotional, behavioural and somatic problems, and poor social integration compared to non-working youth [23].

When considering causal pathways between work and psychosocial harm, a systematic review on child labour and mental health suggested that child workers tended to internalise problems (vs. externalise) and the effects seemed to emerge from isolation (as in domestic work), low self-esteem and a limited sense of control [6]. However, despite work-related stressors, children working in the brick industry had lower levels of anxiety, depression and stronger perceived well-being than the children not in hazardous work, for which the authors highlighted the role of protective factors (e.g., family support and psychological resilience) against the deleterious impact of harmful work conditions [23]. These findings on the harmful effects of isolation, low self-esteem and uncontrollable aspects of events, combined with the protective nature of family support and psychological resilience should be explored to inform intervention designs for children who have been exposed to hazardous work. In this study, hazardous child labour was defined by the number of age-specific work hours and recent exposure (during last week) to a dangerous physical work environment [17]. The measures did not, however, account for psychosocial hazards in the workplace, such as a lack of authority and (physical) power, harsh negligent treatment, discrimination, humiliation, abuse, violence, stigmatisation, lack of secure/supportive relationships, and the deprivation of intellectual and emotional stimuli, which are reported to affect the health and well-being of working children [3,24].

Our analysis also shows the negative effects of various types of child labour on education and schooling. We found a higher probability of school dropout with increasing intensity of child labour; children in hazardous child labour were nearly six times, child labourers were four times and those in hazardous work (without child labour) were nearly two times more likely to drop out than those who had not been involved in any form of child labour. Similarly, a study in Bangladeshi slums examining dropout rates and work indicated that 85% of 6–14-year-old working children worked more than the 42 h weekly legal limit with a majority being out of school [25]. Meanwhile, another MICS analysis of 30 LMICs reported that involvement in child labour significantly reduced school enrolment in 62% of the countries [26]. School plays an important role in the mental health and psychosocial well-being of children by providing intellectual and emotional stimuli, and the chance to form positive meaningful relationships with peers [27]. Well-delivered education promotes children’s cognitive and emotional development and vice versa, as emotionally healthy youth tend to excel in education [28,29,30].

Our subgroup analysis found increased vulnerabilities to hazardous work exposure in school dropouts and children with psychosocial functioning difficulty. Among the school dropouts, children involved in hazardous work were twice as likely to have psychosocial functioning difficulty compared to children neither in child labour nor in a hazardous workplace, while such effects were not evident in the school-attending group. Furthermore, the strength of association between hazardous work and school dropout was relatively stronger in children with psychosocial functioning difficulty (aOR: 2.5) as compared to children without psychosocial functioning difficulty (aOR: 1.5). A study conducted by Pellenq et al. (2021) reported that children working in hazardous conditions in the brick industry were at risk of poor psychosocial well-being. These working youth also had a higher sense of mistreatment, lower level of social support and were deprived of educational opportunities, whereas these effects were apparently lower among non-working children [10]. Additionally, a study from Jordan on a nationally representative sample of children observed that combining school and work caused more harm to the psychosocial well-being of children compared to children who were working but not undergoing schooling [31]. Despite this conflicting evidence, the differential effects of hazardous work by school dropout status suggest indirect connections between hazardous work and psychosocial functioning difficulty, which, in turn, influence school dropout. Alternatively, hazardous work, by affecting school attendance, leads to the poor psychosocial well-being of children. Therefore, it will be important to consider the potential protective effects of socio-emotional learning interventions that can help children to promote their socio-emotional competencies and reduce mental health problems [32]. The promising effects of such programmes have been shown in recent systematic reviews [32,33,34]. Socio-emotionally competent children are likely to succeed academically by fostering and nurturing meaningful relationships with others and acquiring learning, which in turn, contributes to greater chances of success later in life.

Our findings suggest that education-focused interventions should be designed to reflect the characteristics of different target groups, which will make them more effective in retaining working children in schools, bringing back youth who have dropped out, and, ultimately, reducing child labour rates [35]. Children who combine work and school can benefit from social protection schemes or support (e.g., scholarships, school meals, transport vouchers), and youth-focused behavioural interventions and cash transfers for families to keep children, especially girls [9,36]. These initiatives have been proven to increase school attendance and participation, although effects on child labour reduction are mixed or limited [36]. Further, children who leave school can benefit from bridging programs, which help them to enter education, transitional/non-formal education programs and remedial or special education programs [9,36]. When re-entry into mainstream education is not feasible, technical and vocational education (TVET) can serve as a valuable alternative to enable children to gain skills to seek employment [37]. However, it is important to recognise that the school infrastructure, quality of education and contextual factors influence school participation, particularly because poor school environments can push children to leave education to work. In Bangladesh, a poorly resourced public school system, combined with high levels of rural poverty and traditional gender and social norms on child marriage foster school dropout [38].

In addition to poor educational and psychosocial outcomes, like other studies, our study indicates that being from a low socio-economic status and a household with low maternal education are determinants of psychosocial functioning difficulty [29,39,40]. Furthermore, our findings suggest that harsh forms of discipline (either physical punishment or psychological aggression) among children (5–14 year olds) increase the likelihood of poor psychosocial functioning compared to non-violent discipline. The children of parents/guardians who believed that children should be physically punished were more likely to have psychosocial functioning difficulty. These findings were similar to findings from a multi-national study from 17 middle- and low-income countries on the harmful effects of violent/coercive discipline on the behavioural, social and psychological domains of children [41] as well as a study from the Colombo district of Sri Lanka [42]. Experts suggest that corporal/harsh punishment may result in immediate compliance; if it is not accompanied by an explanation, children do not internalise desirable behaviours and this treatment can cause antisocial behaviours and depressive symptoms and may escalate into physical abuse in the long-term [42].

Finally, more research is needed to understand our finding that school dropout was more likely among those who did not experience harsh discipline, as it will be useful to understand how a harsh parenting style, with a focus on ensuring that children stay in school, might have influenced academic performance [43]. In a nationally representative sample of middle school children in China, an authoritative parenting style with high parental involvement and supervision and responsiveness to children’s needs appeared to be associated with educational success [44]. Other studies suggest that predictors of educational achievement include poverty, older student age, poor psychosocial status and parental education, which echoes our result [45,46,47].

### 4.1. Strengths and Limitations

This study draws its strength from the utilisation of a large population-based dataset accessed through a government survey, and the examination of crucial variables such as school dropout, child labour, parental behaviours and the general adjustment of children. Despite a cross-sectional design that prevents us from examining causal relationships, the findings can serve as a foundation for future longitudinal, qualitative and ethnographic research into the exploitation, neglect and maltreatment of children, allowing for the exploration of causal pathways. However, this analysis had several limitations. First, the MICS methodology captures child labour by work activities (paid/unpaid economic activity and house chores), work conditions and age-specific thresholds of work hours [17]. Compared to the SIMPOC (Statistical Information and Monitoring Programme on Child Labour) and LMS (Living Standard Measurement Survey) instruments, MICS’s child labour module is designed to be shorter to minimise survey length and to collect information on multiple other topics and is therefore less comprehensive [48]. This approach did not measure occupational and health risks, remuneration, mistreatment experienced at work, other work-related injury/illness and sexual exploitation that can be disguised in the form of early marriage. The latter is particularly noteworthy in the context of Bangladesh, which has high rates of child marriage in South Asia and cultural uniqueness [49,50]. MICS 6 found out that 60% of women were married before the age of 18 years [17] and that child marriage is a means to force children into commercial sex work and is often associated with a lack of education or a lower education level [49,50]. As with the reported protective effects of secondary education against child marriage in Bangladesh [51], enhancing the education of children can have favourable effects in terms of reducing both child labour and marriage rates. In addition, the underreporting of the child labour and child functioning estimates is likely because the mother/primary caretaker of children was asked rather than the children directly [17].

Second, the variable for psychosocial functioning was constructed from the psychosocial component of the ‘child functioning module’ (CFM) (5–17 years) of the larger Bangladesh MICS excluding hearing, seeing, walking and self-care domains [17,52]. The CFM was pre-tested and compared with other disability measurement tools such as the Washington Group Short Set (WG-SS) and Ten Question Screening Instrument (TQSI), showing it to be a reliable instrument in identifying the functional difficulty of children [17,52,53]. Our results using the 2019 BMICS reported a psychosocial functioning difficulty prevalence of 11% among children in hazardous child labour in Bangladesh, which was comparatively lower than the percentage of behavioural and emotional problems among working children in Ethiopia (20.1%) [54] and Brazil (21.4%) [55]. This difference in reported estimates of psychosocial problems may partly correspond to different measurement instruments used in these studies; Diagnostic Interview for Children and Adolescents (DICA) to categorize psychiatric disorders and child behaviour checklist (CBCL) to capture children’s competencies and problems were used in Ethiopia [54] and Brazil [55], respectively, while UNICEF employed CFM, designed to capture the disability aspect of important functional domains of child development [52]. Even with the same CFM instrument applied, the contextual variation in the estimates of children with functional difficulty was noted. The field testing of CFM in 2014–2016 (including seeing, hearing, walking and self-care domains) elicited a wide variation in the prevalence of functional difficulty among 5–17 year olds in Mexico (46.3%), Samoa (9.3%) and Serbia (25.2%) [52], while the 2019 BMICS estimated that 8.3% of children had functional difficulty with a high prevalence across socio-emotional domains which remain consistent in all four countries [17,52]. Probably, different social norms, cultural interpretations and level of accommodations for disability in these contexts may also influence reported estimates, given that CFM takes scaled responses from caregivers such as “some difficulty” and “a lot of difficulty” to determine the level of difficulty in children.

### 4.2. Implications

There can be little doubt that child labour is undesirable but is a coping economic strategy for poor families against income shocks and in locations where educational options are limited or are costly for families. Reports from around the world indicate reductions in child labour, but suggest that child labour often remains a common feature of low-income settings, despite legislation that prohibits employing children beyond safe hours and to undertake hazardous tasks. Therefore, while eradication should remain a priority, as noted in Target 8.7 of the 2030 Sustainable Development Goals (SDG) that calls for ‘the elimination of the worst forms of child labour …’, those working to assist children exposed to harmful levels and types of work should take account of both the consequences and protective factors to support the mental health and social well-being of working youth. Furthermore, since cognitive and psychosocial problems in childhood are risk factors for psychiatric disorders in adolescents and adults [39,56], initiatives that promote the psychosocial well-being of children should be implemented [17]. Our findings offer evidence that supports the need for educational and social interventions that explicitly promote children’s psychological well-being and address child labour within target groups (e.g., child, family and society and school environments).

## 5. Conclusions

Ultimately, as noted in a recent report by the ILO and UNICEF, the most effective way to address child labour will be to prevent children from engaging in work by supporting their families via social protection schemes such as cash transfer programmes, as well as mechanisms such as universal basic income, universal education and/or universal health insurance [57]. However, until nations decide to support and extend current schemes, those concerned with child protection will need to identify the most feasible and effective ways to protect the health and well-being of working children.

## Figures and Tables

**Figure 1 children-10-01021-f001:**
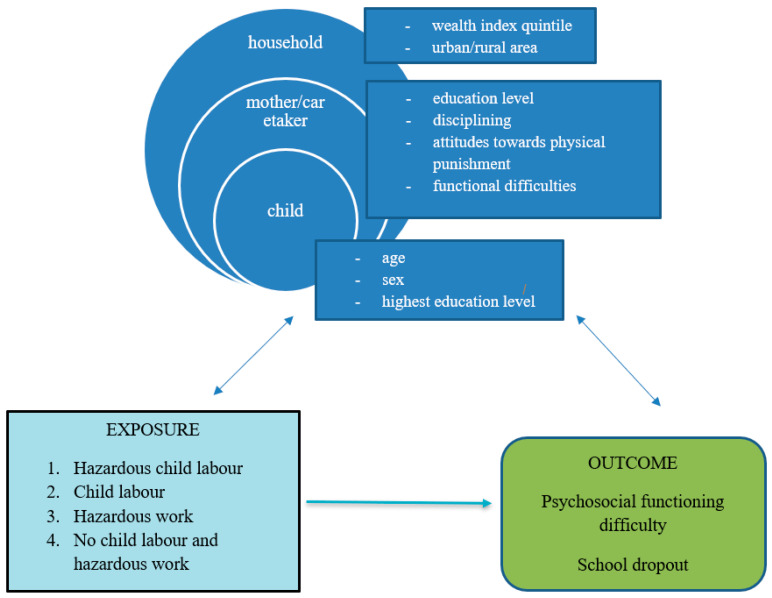
Conceptual framework of associations between hazardous child labour, psychosocial functioning difficulty and school dropout in children in Bangladesh.

**Table 1 children-10-01021-t001:** Operational definitions of the variables.

Exposure Variable	
Hazardous child labour consists of four categories	
Child labour [17]	Children were classified as being in child labour if they performed economic activities or household chores during the last week above the age-specific cut-off points. This included paid or unpaid work for someone outside their immediate family or work for a family farm or business, or household chores, such as cooking, cleaning or caring for children, as well as collecting firewood or fetching water. For economic activities, the age-specific numbers of hours are as follows: (i) age 5–11: ≥1 h; (ii) age 12–14: ≥14 h; (iii) age 15–17: ≥43 h. For household chores, the age-specific numbers of hours are as follows: (i). age 5–11 and age 12–14: ≥21 h; (ii) age 15–17: ≥43 h [17].
Hazardous work [17]	Children were considered to be in hazardous work if they performed/worked under any of these conditions: (i) carrying heavy loads; (ii) working with dangerous tools or operating heavy machinery; (iii) exposed to dust, fumes or gas; (iv) exposed to extreme cold, heat or humidity; (v) exposed to loud noise or vibration; (vi) working at heights; (vii) working with chemicals or explosives; or (viii) exposed to other unsafe or unhealthy processes or conditions. Children in this group do not meet the UNICEF criteria of child labour defined by age-specific thresholds of work hours for economic or household chores.
Hazardous child labour [17]	Children engaged in child labour and who were in hazardous work were classified as hazardous child labour.
No child labour or hazardous work	Children defined as not being in child labour and hazardous work when they were involved in an economic or household activity below age-specific thresholds of child labour and performed no hazardous work.
Outcome variable	
Psychosocial functioning difficulty consists of two categories	
Yes	Children with difficulty in at least one of the following domains: communication, learning, remembering, concentration, accepting change, controlling behaviour, making friends, anxiety and depression.
No	Children without psychosocial functioning difficulty
School dropout consists of two categories	
Yes [17]	Children attending school in the past year of the survey and not currently attending school at the time of the survey
No [17]	Children attending standard school in the past year and in the current year of the survey.
Confounding variable	
Discipline consists of three categories	
Violent discipline [17]	1.1. Psychological aggression includes either one of the acts of (i) shouting, yelling or screaming at a child; or (ii) calling a child offensive names, such as ‘dumb’ or ‘lazy’. 1.2. Physical punishment includes either one of the acts of (i) shaking the child; (ii) hitting or slapping him/her on the face, head or ears; (iii) hitting or slapping him/her on the hand/arm/leg; (iv) hitting him/her on the bottom or elsewhere on the body with a hard object; (v) spanking or hitting him/her on the bottom with a bare hand; or (vi) beating him/her over and over as hard as possible
Non-violent discipline [17]	Consists of either one of the following: (i) explaining why a behaviour is wrong; (ii) taking away privileges or not allowing him/her to leave the house; or (iii) giving him/her something else to do.
No discipline	When they answered ‘no’ to any of the above disciplinary acts.

**Table 2 children-10-01021-t002:** Characteristics of children aged 5–17 years in Bangladesh MICS in 2019.

	Total N (%)
Total	66,705 (100)
Age of child at the time of survey	
5–11 yrs.	35,505 (53.23)
12–14 yrs.	16,007 (24)
15–17 yrs.	15,193 (22.78)
Sex	
Male	33,901 (50.82)
Female	32,803 (49.18)
Wealth index quintile	
Poorest	14,693 (22.03)
Second	14,239 (21.35)
Middle	13,176 (19.75)
Fourth	12,348 (18.51)
Richest	12,249 (18.36)
Area	
Urban	13,664 (20.48)
Rural	53,041 (79.52)
Mother’s education	
Pre-primary or none	18,216 (27.31)
Primary	19,155 (28.72)
Secondary	24,411 (36.6)
Higher secondary and above	4923 (7.38)
Child’s highest level of education attended	
ECE	5548 (8.72)
PRIMARY	31,559 (49.6)
Secondary	15,485 (24.34)
Secondary/Higher secondary	10,996 (17.28)
Higher	37 (0.06)
No response	0
Child’s psychosocial functioning difficulty	
No	61,765 (92.59)
Yes	4940 (7.41)
School dropout	
No	55,730 (88.2)
Yes	7457 (11.8)
Hazardous work conditions	
No	61,388 (92.03)
Yes	5316 (7.97)
Child labour	
Below age thresholds	35,148 (53.48)
At or above age thresholds	4461 (6.79)
Hazardous child labour	
No	26,107 (39.73)
No child labour and hazardous work	59,195 (88.74)
Child labour	2194 (3.29)
Hazardous child labour	2267 (3.4)
Hazardous work	3049 (4.57)
Discipline *	
Non-violent discipline	3613 (7.3)
Violent discipline	45,863 (92.7)
No discipline	1 (0)
Parent/Guardian perception: “Child needs to be physically punished to be brought up properly”	
Yes	13,762 (35.09)
No	25,288 (64.48)
DK/No opinion	129 (0.33)
No response	39 (0.1)
Caregiver’s functional difficulties (age 18–49 years)	
Has functional difficulty	1968 (2.95)
Has no functional difficulty	57,012 (85.47)
No information	7724 (11.58)

* denominator of “discipline” variable was available only for children aged 5–14 years (*n* = 49,477), n.a. = not applicable, household work activities are collected only for 5–14 years.

**Table 3 children-10-01021-t003:** Hazardous child labour and psychosocial functioning difficulty in children aged 5–14 years in Bangladesh MICS (N = 20,388).

	Crude OR (95% CI)	*p*-Value	Adjusted OR (95% CI)	*p*-Value
Age of child at the time of survey				
	0.98 (0.97–0.98)	0.000	0.98 (0.96–1.00)	0.105
Sex				
Male	Ref		Ref	
Female	**0.84 (0.80–0.89)**	**0.000**	**0.90 (0.83–0.97)**	**0.009**
Wealth index quintile				
Poorest	Ref		Ref	
Second	0.98 (0.90–1.06)	0.578	1.02 (0.91–1.14)	0.778
Middle	**0.82 (0.75–0.89)**	**0.000**	**0.82 (0.73–0.93)**	**0.002**
Fourth	**0.69 (0.63–0.75)**	**0.000**	**0.71 (0.61–0.81)**	**0.000**
Richest	**0.60 (0.54–0.66)**	**0.000**	**0.74 (0.63–0.87)**	**0.000**
Area				
Urban	Ref		Ref	
Rural	**1.43 (1.32–1.55)**	**0.000**	**1.25 (1.11–1.42)**	**0.000**
Mother’s education				
Pre-primary or none	Ref		Ref	
Primary	0.92 (0.85–0.99)	0.033	1.08 (0.97–1.21)	0.147
Secondary	**0.86 (0.80–0.93)**	**0.000**	1.06 (0.94–1.18)	0.352
Higher secondary and above	**0.76 (0.67–0.86)**	**0.000**	1.19 (0.98–1.45)	0.081
Child’s highest level of education attended				
ECE	1	Ref	1	Ref
Primary	0.98 (0.88–1.10)	0.780	1.00 (0.86–1.17)	0.147
Secondary	**0.85 (0.75–0.95)**	**0.005**	0.98 (0.79–1.22)	0.352
Secondary/higher secondary	**0.66 (0.58–0.75)**	**0.000**	0.85 (0.62–1.17)	0.081
School dropout				
No	1	Ref	1	Ref
Yes	**1.32 (1.21–1.44)**	**0.000**	**1.63 (1.39–1.92)**	**0.000**
Hazardous child labour				
No child labour and hazardous work	1	Ref	1	Ref
Child labour	**1.3 (1.12–1.51)**	**0.001**	1.08 (0.88–1.32)	0.471
Hazardous child labour	**1.61 (1.4–1.84)**	**0.000**	**1.41 (1.16–1.72)**	**0.001**
Hazardous work	**1.2 (1.06–1.37)**	**0.005**	**1.39 (1.1–1.76)**	**0.006**
Discipline				
Non-violent discipline	1	Ref	1	Ref
Violent discipline	**1.72 (1.47–2.01)**	**0.000**	**1.73 (1.43–2.10)**	**0.000**
Parent/Guardian perception: “Child needs to be physically punished to be brought up properly”				
Yes	1	Ref	1	Ref
No	**0.58 (0.54–0.63)**	**0.000**	**0.61 (0.56–0.66)**	**0.000**
DK/No opinion	1.32 (0.79–2.21)	0.288	1.47 (0.85–2.52)	0.166
No response	1.85 (0.81–4.24)	0.146	2.02 (0.77–5.33)	0.155
Caregiver’s functional difficulties (age 18–49 years) (*n* = 39,386)				
Has functional difficulty	1	Ref	1	Ref
Has no functional difficulty	**0.25 (0.23–0.28)**	**0.000**	**0.26 (0.22–0.30)**	**0.000**
No information	**0.27 (0.23–0.31)**	**0.000**	**0.31 (0.25–0.37)**	**0.000**

OR—odds ratio. Crude and adjusted OR results with *p* value ≤ 0.05 are shown in bold.

**Table 4 children-10-01021-t004:** * Association between Hazardous Child Labour and Psychosocial Functioning Difficulty by School Dropouts among Children aged 5–14 years in Bangladesh MICS.

	School Dropout (N = 1221)		No School Dropout (N = 19,393)	
	Adjusted OR (95% CI)	*p*-Value	Adjusted OR (95% CI)	*p*-Value
Hazardous child labour				
No child labour and hazardous work	1	Ref	1	Ref
Child labour	1.12 (0.71–1.78)	0.622	1.09 (0.88–1.38)	0.412
Hazardous child labour	**1.48 (1.02–2.15)**	**0.038**	**1.52 (1.2–1.93)**	**0.000**
Hazardous work	**2.55 (1.61–4.06)**	**0.000**	1.11 (0.84–1.49)	0.451

* Logistic regression models were run in two groups: school dropouts and school attending group, accounting for potential confounders included in Table 3 to assess the relationship between hazardous child labour and psychosocial functioning difficulty. Results are only shown for hazardous child labour. OR—odds ratio. Crude and adjusted OR results with *p* value ≤ 0.05 are shown in bold.

**Table 5 children-10-01021-t005:** Hazardous child labour and school dropout in children aged 5–14 years in Bangladesh MICS (N = 20,339).

	Crude OR (95% CI)	*p*-Value	Adjusted OR (95% CI)	*p*-Value
Age of child at the time of survey				
	**1.49 (1.48–1.51)**	**0.000**	**1.51 (1.47–1.54)**	**0.000**
Sex				
Male	1	Ref		
Female	**0.54 (0.51–0.56)**	**0.000**	**0.38 (0.35–0.43)**	**0.000**
Wealth index quintile				
Poorest	1	Ref		
Second	**0.88 (0.82–0.94)**	**0.000**	**0.86 (0.76–0.98)**	**0.027**
Middle	**0.76 (0.70–0.81)**	**0.000**	**0.86 (0.75- 0.99)**	**0.037**
Fourth	**0.65 (0.60–0.70)**	**0.000**	**0.83 (0.72–0.97)**	**0.021**
Richest	**0.44 (0.40–0.48)**	**0.000**	**0.67 (0.55–0.82)**	**0.000**
Area				
Urban	1	Ref		
Rural	**1.07 (1.01–1.14)**	**0.026**	**0.63 (0.56–0.72)**	**0.000**
Mother’s education				
Pre-primary or none	1	Ref		
Primary	**0.56 (0.53–0.59)**	**0.000**	**0.68 (0.60–0.76)**	**0.000**
Secondary	**0.26 (0.25–0.28)**	**0.000**	**0.34 (0.30–0.39)**	**0.000**
Higher secondary and above	**0.09 (0.08–0.11)**	**0.000**	**0.21 (0.15–0.29)**	**0.000**
Child’s psychosocial functioning difficulty				
No	1	Ref		
Yes	**1.32 (1.21–1.44)**	**0.000**	**1.71 (1.46–1.99)**	**0.000**
Hazardous child labour				
No child labour and hazardous work	1	Ref	1	Ref
Child labour	**4.67 (4.23–5.16)**	**0.000**	**3.97 (3.38–4.67)**	**0.000**
Hazardous child labour	**12.9 (11.85–14.23)**	**0.000**	**5.65 (4.83–6.61)**	**0.000**
Hazardous work	**5.71 (5.26–6.2)**	**0.000**		**0.000**
Discipline				
Non-violent discipline	1	Ref		
Violent discipline	**0.47 (0.42–0.53)**	**0.000**	**0.68 (0.59–0.79)**	**0.000**
Parent/Guardian perception: “Child needs to be physically punished to be brought up properly”				
Yes	1	Ref		
No	1.05 (0.96–1.15)	0.255	1.07 (0.97–1.19)	0.171
DK/No opinion	1.65 (0.90–3.01)	0.105	0.67 (0.28–1.59)	0.363
No response	0.90 (0.22–3.64)	0.886	1.16 (0.26–5.19)	0.847
Caregiver’s functional difficulties (age 18–49 years)				
Has functional difficulty	1	Ref		
Has no functional difficulty	**0.67 (0.59–0.77)**	**0.000**	0.98 (0.77–1.26)	0.883
No information	**2.57 (2.23–2.96)**	**0.000**	1.25 (0.96–1.63)	0.100

Violent variables: violent/non-violent disciplines and attitudes towards physical punishment are only for age 5–14 years. OR—odds ratio. Crude and adjusted OR results with *p* value ≤ 0.05 are shown in bold.

**Table 6 children-10-01021-t006:** * Association between Hazardous Child Labour and School Dropouts by Psychosocial Functioning Difficulty among Children aged 5–14 years in Bangladesh MICS.

	Psychosocial Functioning Difficulty (N = 1219)		No Psychosocial Functioning Difficulty (N = 19,007)	
	Adjusted OR (95% CI)	*p*-Value	Adjusted OR (95% CI)	*p*-Value
Hazardous child labour				
No child labour and hazardous work	1	Ref	1	Ref
Child labour	**3.33 (1.97–5.64)**	**0.000**	**4.03 (3.4–4.78)**	**0.000**
Hazardous child labour	**4.50 (2.9–6.99)**	**0.000**	**5.81 (4.91–6.87)**	**0.000**
Hazardous work	**2.52 (1.47–4.34)**	**0.001**	**1.54 (1.23–1.92)**	**0.000**

* Logistic regression models were run in two groups: children with psychosocial functioning difficulty and children without psychosocial functioning difficulty, accounting for confounders included in Table 5 to assess the relationship between hazardous child labour and school dropout. Results are only shown for hazardous child labour. OR—odds ratio. Crude and adjusted OR results with *p* value ≤ 0.05 are shown in bold.

## Data Availability

The data presented in this study are openly available at https://mics.unicef.org/, accessed on 21 July 2021.

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
