# Peer review of "Hazardous Child Labour, Psychosocial Functioning, and School Dropouts among Children in Bangladesh: A Cross-Sectional Analysis of UNICEF’s Multiple Indicator Cluster Surveys (MICS)"

_children, 2023, doi:10.3390/children10061021_

Round 1

Reviewer 1 Report

The paper is well written and presented, but the findings of associated factors do not add new content to the literature.

Below are my questions and comments.

Limitation (information bias): the authors state that the mothers/caretakers of the sampled children were the respondents of the questionnaire.  Is the psychosocial functioning instrument for children validated for the responses of  mothers/caretakers? How reliable is the estimated prevalence of psychosocial functioning? How reliable is the child labour variable reported by the mothers/caretakers?

Are the instruments used in others countries comparable to those used in Bangladesh? Do they have the same strenghts and weaknesses?

It is necesssary to report that the prevalence of outcomes may be underestimated.

Reviewer 2 Report

The strength of this study is that it uses a large sample of families, accessed through a government survey, of the important variables of school dropout, child labor, parental behaviors, and child's general adjustment. Since this a study at one point in time, causal relations between these variables cannot be inferred, but the information offered does provide the basis for further longitudinal, qualitative and ethnographic studies of the exploitation, neglect  and maltreatment of children in which causal pathways may be explored. This might be added to the discussion.

I would also like to see, added to the discussion, the acknowledgement of two forms of child labour which the study design has failed to consider. The first is that of imposing on young teenaged girls the 'labor' of housework and sexual service through early marriage, which forces school dropout. Available statistics suggest that rates of marriage in girls of school age in Bangladesh are amongst the highest in Asia. 

A second form of child labor which has profoundly negative effects on school-aged girls is that of enforced prostitution. Enforced prostitution of Bangladeshi girls has a certain cultural uniqueness in this country  - see reference below. 

Bagley, C. , Kadri, S. , Shahnaz, A. , Simkhada, P. and King, K. (2017) Commercialised Sexual Exploitation of Children, Adolescents and Women: Health and Social Structure in Bangladesh. Advances in Applied Sociology7, 137-150. doi: 10.4236/aasoci.2017.74008.

Reviewer 3 Report

First of all, thank you for you the interesting paper. Here are some of my comments and suggestions:

-       Outcome variable: please specify your outcome variable(s) and stay consistent as you mentioned different constructs in different places: 

o   child labor/psychosocial difficulty/ school drop-out (LL: 130). 

o   Table 1: specifies both psychosocial difficulty and drop-outs under psychosocial functioning. 

o   “multivariable logistic regression analyses were adjusted for potential confounding variables to determine the relationship between hazardous child labour, psychosocial functioning and then the relationship between hazardous child labour with school dropout and psychosocial functioning. (LL.163-167)

o   The Table 3 is titled as “Table 3Hazardous child labour and psychosocial functioning of children aged 5-17”.

o   “Hazardous child labour (AOR:1.41; 95% CI:1.16-1.72; P: 0.001) and hazardous work (AOR: 1.39; 95% CI: 1.1-1.76; P: 0.006) were associated psychosocial dysfunction, after adjusting for confounders in 5–14-year-olds (LL: 197-198)

-       Results: You have represented results for different ages, however, the title of Table 3 indicates that it represents findings from all 5-17-year olds. Did you run separate models for each age category? Please ensure that your tables and narratives correspond to and complement each other. The same applies to Table 4.

-       Discussion: you argue that “Our analysis also contributes to evidence on school dropout and psychosocial harm, 288 suggesting that school modifies the correlation between hazardous child labour and psychosocial functioning difficulty in children” (LL: 288-290). Please provide more details on how did you test this in your models.

Reviewer 4 Report

This review is inadequate due to its unclear presentation.

The objective is ambiguous, and only the adverse consequences of child labor are emphasized, without any discussion of potential solutions to address the issue.

The methods employed should have been clarified and elaborated upon to enhance comprehension.

The outcomes are challenging to interpret, and the tables should be simplified and streamlined for ease of use.

There is a dearth of information on future prospects for reducing child labor rates.

Furthermore, the bibliography is outdated and requires updating with more recent sources.

Round 2

Reviewer 4 Report

The work is now much clearer as all the mentioned critical points have been addressed satisfactorily. The introduction has been anhanced by incorporating a new and updated bibliography. Additionally, the methods and results are presented in a reader - friendly manner